# A Computational Modeling and Simulation Workflow to Investigate the Impact of Patient-Specific and Device Factors on Hemodynamic Measurements from Non-Invasive Photoplethysmography

**DOI:** 10.3390/bios12080598

**Published:** 2022-08-04

**Authors:** Jesse Fine, Michael J. McShane, Gerard L. Coté, Christopher G. Scully

**Affiliations:** 1Department of Biomedical Engineering, Texas A&M University, College Station, TX 77843, USA; 2Center for Remote Health Technologies and Systems, Texas A&M Engineering Experiment Station, Texas A&M University, College Station, TX 77843, USA; 3Department of Materials Science and Engineering, Texas A&M University, College Station, TX 77843, USA; 4Office of Science and Engineering Laboratories, Division of Biomedical Physics, Center for Devices and Radiological Health, Food and Drug Administration, Silver Spring, MD 20993, USA

**Keywords:** photoplethysmography, remote monitoring, computational modeling and simulation, medical device design

## Abstract

Cardiovascular disease is the leading cause of death globally. To provide continuous monitoring of blood pressure (BP), a parameter which has shown to improve health outcomes when monitored closely, many groups are trying to measure blood pressure via noninvasive photoplethysmography (PPG). However, the PPG waveform is subject to variation as a function of patient-specific and device factors and thus a platform to enable the evaluation of these factors on the PPG waveform and subsequent hemodynamic parameter prediction would enable device development. Here, we present a computational workflow that combines Monte Carlo modeling (MC), gaussian combination, and additive noise to create synthetic dataset of volar fingertip PPG waveforms representative of a diverse cohort. First, MC is used to determine PPG amplitude across age, skin tone, and device wavelength. Then, gaussian combination generates accurate PPG waveforms, and signal processing enables data filtration and feature extraction. We improve the limitations of current synthetic PPG frameworks by enabling inclusion of physiological and anatomical effects from body site, skin tone, and age. We then show how the datasets can be used to examine effects of device characteristics such as wavelength, analog to digital converter specifications, filtering method, and feature extraction. Lastly, we demonstrate the use of this framework to show the insensitivity of a support vector machine predictive algorithm compared to a neural network and bagged trees algorithm.

## 1. Introduction

Cardiovascular disease (CVD) is the leading cause of death globally, with an estimated 17.9 million people dying from CVD in 2019 [1]. Of these deaths nearly 85% were due to heart attack or stroke [1]. Nearly half of all adults in the United States (116 million or 47%) have hypertension, commonly referred to as high blood pressure [2]. Studies show that every 10 mmHg drop in systolic BP reduces the probability of heart attack and stroke by ~50% for all age groups [3]. Lowering systolic BP from 140 to 120 mmHg has also shown to reduce the risk of death by ~27% [4]. Thus, monitoring blood pressure to identify and subsequently address hypertension is a common and effective way to reduce risk of developing CVD [5]. Monitoring these parameters noninvasively and continuously could provide additional insight into a patient’s blood pressure over time to enable earlier detection and improved management of hypertension [6,7,8]. Noninvasive and continuous methods to monitor blood pressure are thus of great interest to the healthcare community to provide care to individuals with CVD or those at-risk of developing CVD.

Photoplethysmography (PPG) is a non-invasive optical technique that has been extensively studied for its potential to non-invasively monitor blood pressure [9,10]. By illuminating the skin and recording the light that reaches nearby photodetectors, the PPG waveform resulting from diffusely reflected photons that interact with blood and tissue are collected. The PPG waveform has a large “quasi-DC” component from static absorption and scattering from tissue and blood, as well as a much smaller “AC” component that is of great interest because it correlates to the pulsatile local blood volume dynamics of microvasculature and/or macrovasculature—depending on the device wavelength and target anatomy [11].

The PPG waveform is most often collected with a sampling frequency between ~120–1000 Hz, which enables Pulse Wave Analysis (PWA) of the AC component [12,13]. Through PWA, the PPG waveform is processed through feature extraction to derive the values of specific characteristics of the waveform. Some PPG waveform features commonly studied are listed in Table 1, and an example waveform with labeled systolic peak, dicrotic notch, and pulse onset is shown in Figure 1. These features are studied as inputs to models that estimate cardiovascular parameters such as blood pressure [14]. However, the PPG waveform is subject to added variability and uncertainty from numerous sources. These factors originate from patient physiology as well as environmental factors and device designs [11]. For similar PPG-like optical absorption measurements such as SpO2, differences in measurement performance has been observed between patient skin tones [15,16]. Additionally, the changes in skin thickness and vessel compliance that accompany age also affect the PPG waveform and signal quality [11]. Other factors such as device wavelength, device analog-to-digital converter specifications, and filtering methodology can alter the recorded PPG waveform and manipulate device signal quality [17,18], potentially impacting if algorithms developed from one device are applicable to another. Thus, as efforts continue to utilize PPG for hemodynamic monitoring applications, quantification of effects of age-related changes, patient skin-tone, and device designs on PPG features can improve robustness and generalizability. Additionally, resources that enable systematic assessment of algorithmic performance on data that are inclusive of these factors could provide more robust design of such devices.

While available clinical monitoring databases such as the MIMIC-III Waveform Database enable testing of algorithms on data from tens of thousands of patients, insufficient data on patient characteristics such as skin tone can limit the utility to understanding specific factors that may affect the performance of a new measurements [20,21]. Additionally, important information on the devices used during clinical monitoring and characteristics of those devices such as signal filter type and schematics of the light emitting diode (LED) and photodiode (PD) may be unknown. Thus, the data available to support algorithm development and how they can be applied to different device designs are limited. Computational modeling is a powerful tool to fill this gap as any permutation of factors can be incorporated and explored.

Monte Carlo modeling (MC) of the optical path and statistical modeling of waveform morphology (e.g., through Gaussian combinations) are tools that have been used to evaluate the effects of physiology or device parameters on raw PPG signal strength [22,23,24]. Chatterjee et al. have used MC for this purpose [25]. MC results from Oxygen saturation simulations at 660 nm and 880 nm across skin tone and source/detector separation were compared to experimentally collected data, the model was validated and the origin of the PPG signal was explored [24,25,26]. Boonya-ananta et al. focused on individuals with dark skin tone and obesity, using MC to generate a single PPG period and then quantified the impact those features have on PPG AC/DC ratio [22].

Despite such advances, it is noteworthy that the use of MC has not yet been extended to evaluate the downstream effects these factors have on algorithm performance or clinical action. One reason why this is the case is because computational time prevents large windows of data from being created. Further, MC for photon propagation is inherently limited in its inability to represent other necessary factors such as motion artifacts that do not manifest in simulation geometry or optical properties. Gaussian combination is a previously reported methodology to generate PPG waveforms over a larger window of time [23]. Tang et al. improved on this work by publishing “PPGSynth” in 2020: a toolbox that uses Gaussian combination to generate arrhythmic PPG signals with a specified sampling frequency, length, heart rate, and noise with an easy-to-use user interface [27]. Gaussian combination, however, is limited by its inability to represent the effect that physiological/anatomical factors have on the PPG waveform. Thus, both of these techniques face limitations to simulate PPG waveforms across populations considering different device designs.

In this work, we combine MC and Gaussian combination to create a platform to assess the impact of device wavelength, device analog to digital converter (ADC) resolution, patient age, and patient skin-tone against PPG feature extraction. By leveraging the ability of MC to represent patient-specific factors and the ability of gaussian combination to generate entire waveforms without utilizing extensive computational resources, we propose that the combination of these two methodologies will enable creation of physiologically and anatomically accurate waveforms to evaluate patient and device specific effects on the PPG waveform. The main purpose of the simulation tool is to generate databases of synthetic PPG signals with customizable parameters that represent a diverse cohort. This will enable end users who are currently developing blood pressure predictive algorithms to use this tool as a cost-effective and accessible methodology to test the robustness of algorithms to these factors. Additionally, as an example test case of this framework, synthesized data are passed through feature extraction tools to study how the patient and device factors impact the waveform characteristics. We then present a case example with different machine learning models (Neural Network, Bagged Trees algorithm, Support Vector Machine) trained to estimate blood pressure from PPG waveforms to show how such an approach can be used to evaluate hemodynamic measurement performance changes as a function of the aforementioned parameters.

## 2. Materials and Methods

### 2.1. Schema Overview

This workflow, conceptually illustrated in Figure 2, involves creating and processing synthetic PPG waveforms with two primary components: the PPG signal generator and the device algorithm simulator. Small icons indicate where a process (such as filtering) or parameter (such as age or skin tone) is relevant. The PPG signal generator creates a 30 s window of data with a sampling frequency of 120 Hz that is representative of a PPG derived from an individual of a specified skin tone and age and taken with a device of a specified wavelength. The PPG signal generator has two sub-components: a Monte Carlo (MC) model and a PPG waveform generator. The MC model estimates the AC and DC components of a PPG signal derived from a combination of the above three factors by simulating photon propagation through tissue, where the dimensions of the tissue and its optical properties change as a function of the age, skin tone, and wavelength being simulated. These AC and DC values are used as inputs into the PPG waveform generator, which uses Gaussian combination and a generalized reduced gradient (GRG) solver to create a waveform that is very similar to an input PPG wave shape from a template pulse. By using Gaussian combination, the end user is able to explore any interpolated PPG waveform morphology. PPG noise can be added to create PPG waveforms under various conditions. Within the device algorithm simulator, these data are filtered, segmented into 30 s windows, and re-scaled to be passed through feature extraction where fiducial points are calculated and used as inputs for blood pressure prediction algorithms. In the current study, device filters are applied to the generated PPG waveforms.

### 2.2. PPG Signal Generator

#### 2.2.1. Monte Carlo Model

A six-layer MC model of the volar fingertip consisting of the epidermis, papillary dermis, upper blood net dermis, reticular dermis, deep blood net dermis, and subcutaneous fat was constructed using MCmatlab (Figure 3, isometric view) [24,28,29]. MCmatlab was chosen due to its ability to simply manipulate geometrical and optical properties of the models developed, as well as ease of use by someone not intimately familiar with MATLAB or Monte Carlo. The isometric view in Figure 3 provides a graphical illustration of the multi-layered model and a legend that names each layer based on the color used. The geometrical and optical properties of the model were dependent on the simulated device wavelength, patient age, and patient skin tone; and are listed in Table 2 [30,31,32,33,34,35]. A cardiac pulse is simulated in the Monte Carlo model by completing two simulations for a given combination of parameters: a first simulation where the optical properties are representative of tissue at rest (“rest”), and a second simulation where the optical properties of the dermal layer are changed to represent an increased blood volume (“pulse”). The result of the first simulation is the DC value, and the result of the second simulation is the AC + DC.

Skin layer thickness are derived from previously published literature [31,32,33]. First, the epidermal thickness was found in literature to be 0.055 cm and the total dermal thickness for a control cohort (termed “healthy”) within a systemic sclerosis study of 61-year-old subjects (4 male, 33 female) was found to be 0.075 cm [32,33]. Then, using data collected by Shuster et al., a line of best fit correlating age to dermal thickness allows estimation of appropriate dermal thickness for subjects of any age. In this work, the line of best fit yielded an annual average decrease of 0.00044 mm [31]. Of the entire dermal thickness, the papillary dermis is 8.2%, the upper blood net dermis is 4.37%, the reticular dermis is 81.97%, and the deep blood net is 5.46% [28]. From this, the thickness of each sublayer at the simulated ages are calculated and can be seen in Table 2.

The Monte Carlo model layers are derived from previous literature: Meglinski et al. created a four-layer dermis for MC modeling, and modifying the blood volume fraction to represent the systolic peak of a PPG waveform has been done previously by Fine et al. and Chatterjee et al. [26,28,36]. Additionally, the optical properties of these layers are derived from Jacques et al. [30]. The model presented herein differs from previous literature by being the first to incorporate patient age by varying dermal layer thickness and vessel compliance.

Many tissue optical properties are included natively in MCmatlab (as derived from Jacques et al.) [29,30]. The epidermal absorption coefficient is changed to account for volume fraction melanosomes (VFM) on the volar fingertip that for this work we used the range from 0.03 to 0.30, shown to be within physiological range for other body sites [37]. The range 0.03 to 0.30 was chosen for this work instead of the typical 0.03 to 0.43 because fingertips have less melanin than most body sites; however, it is a limitation that there is not a direct reference for experimental values at this site. Additionally, the default MCmatlab subcutaneous fat absorption coefficient is multiplied by 0.10 to be respective of values found in literature [38].

In the case of the pulse Monte Carlo model, the absorption coefficient of the dermal sublayers is increased to represent an additional absorptive contribution by oxygenated blood. Modifying the magnitude of increase also allows simulating changes in vessel compliance that accompany age, as the volar fingertip is vascularized by a subungual arcade instead of a discrete artery. The increase in blood volume is calculated based on values from literature. The change in carotid artery diameter during the cardiac cycle across ages was converted to a change in area [34]. Next, the change in area of the carotid artery as a function of age was assumed to be constant for the largest artery supplying the subungual arcade of the fingertip- the proper digital artery, which increases in diameter 6% and has a cross sectional area change of 12% for a healthy ~20 year old subject [39]. Finally, it was estimated that the increase in blood volume at the systolic peak is approximately 1.124× for a 23-year-old, 1.099× for a 34.4-year-old, 1.083× for a 44.8-year-old, and 1.073× for a 55-year-old.

For this study, the Monte Carlo model used a square LED-type emitter (top-hat distribution in the near field and Lambertian distribution in the far field) with a side length of 1.0 mm and a half angle of 2.4 radians. The photodiode is donut-shaped surrounding the emitter with an inner radius of 0.0071 mm and an outer radius of 0.0091 mm. However, photodiode and LED size, half angles, and distributions can be changed as needed for the end user. These use-case values were chosen to be similar to the existing Apple Watch LED/PD configuration.

The simulations were completed on a PowerSpec G900 (Micro Electronics Inc, Hilliard, OH, USA) and were parallelized to the GPU: an NVIDIA GeForce RTX 3070 8 GB (NVIDIA Corporation, Santa Clara, CA, USA). To determine the number of photons required for each simulation, every simulation was completed in triplicate (a given combination of parameters required 3 rest simulations and 3 pulse simulations) and repeated with an increasing number of photons until the coefficient of variation of the AC across the triplicate results was less than 10%. The AC is defined as the result of the pulse simulation minus the result of the rest simulation, whereas the DC is the result of the rest simulation. This value was found to be dependent on the parameters of a given simulation, ranging from 5 × 10^8^ photons for the case of device wavelength of 515 nm and volume fraction melanosomes 0.03 to 1 × 10^11^ photons in the case of device wavelength of 880 nm and volume fraction melanosomes 0.30. In total, 96 simulations were completed at the sufficient number of photons, requiring approximately 3.5 weeks.

The outputs of these simulations are synthetic AC and DC amplitudes. To enable comparison of AC values across wavelength, age, and skin tone as well as assess the simulated results with respect to other works; the AC values were normalized to the maximum value such that the largest AC value has a normalized value of 1. Lastly, a blue/white colormap was applied to enable qualitative comparisons such that the maximum value is blue and the minimum value is white. The same procedure was applied to the DC amplitudes.

#### 2.2.2. PPG Waveform Generator

The PPG waveform generator creates a continuous and synthetic PPG waveform intended to represent a combination of patient- and device- specific factors. For this work, the shape of a single waveform period was derived from Allen and Murray, which details the average volar fingertip PPG waveform from individuals that are 23 years-old (YO), 34.4 YO, 44.8 YO, and 55 YO [40]. These four waveforms were chosen as they represent morphologically different waveforms observed in PPG data analysis: namely, loss of diastolic peak and dicrotic notch. Other waveforms were not included to maintain a reasonable length work and limit time needed to run all simulations. To generate full waveforms representative of patients with various ages for the study, the PPG Waveform Generator utilizes an input waveform collected from the volar fingertip with an infra-red LED found in literature [41]. A generalized reduced gradient (GRG) solver native to Microsoft excel was used to determine the amplitude (a1, a2, a3), bias (b1, b2, b3) and width (c1, c2, c3) of three Gaussians that additively combine and minimize relative error when compared to the input waveforms from the literature, and enables creation of PPG waveforms by the end user not included within this work.

After a single 0.8 s period of the PPG waveform is created, a heart rate (75 beats per minute), and sampling frequency (120 Hz) are specified and the waveform is repeated for a specified signal length. Finally, any second waveform or noise, including noise from real PPG data, can be added to the waveform to imitate noise. Here, the noise waveform is constructed via adding sinusoids of customizable frequencies and relative amplitudes to mimic common noise frequencies for PPG signals. Specifically, in this work, three noise sinusoids at 20 Hz, 40 Hz, and 60 Hz and amplitudes 38%, 59%, and 59% of the AC amplitude were used, as measured in previous literature [42].

To analyze the accuracy of the waveform generator, one period of the resultant waveform for each of the four ages is compared to the waveform derived from literature [41]. Median relative error was the chosen statistical measure to make this comparison, as it provides the ability to compare difference between the input and output waveforms while also being robust to larger relative differences at the tail ends of the waveforms that would have their relative difference overrepresented with other measures, such as mean relative error.

### 2.3. Device Algorithm Simulator

#### 2.3.1. Signal Preprocessing

The main components of signal preprocessing simulated here includes high level components: filtering, ADC simulation, and rescaling. The data is filtered by one of the following: a 0.1–7 Hz 4th order Butterworth Bandpass, a 0.1–7 Hz 4th order Inverse Chebyshev bandpass with a 10-element moving average filter, and a 7 Hz 4th order Inverse Chebyshev lowpass with a 10-element moving average filter. ADC simulation is completed by allowing the user to specify an ADC resolution in bits, a reference current in amps, and LED power in Watts. The ADC value is found by using Equation (1):ADC Values = MC_Out × (2^ADC Resolution)/Reference Current(1)
where MC_Out is the output AC of the MC simulation modified by:MC_Out = 10 × AC × LED power × photodiode area(2)

The morphology of the PPG waveforms with the smallest and largest AC amplitudes are evaluated across ADC values. First, an assumed reference current of 32 uA and an assumed LED power of 50 mW are used. PPG waveform data is simulated with an ADC from 1 bit to 25 bits for the patient-specific and device factors that yield the highest and lowest AC amplitudes. Next, waveforms collected with select simulated ADC resolutions are qualitatively compared to determine the impact that insufficient ADC resolution can have on PPG morphology. The percent difference between a PPG waveform at a given ADC resolution and the PPG waveform at an ADC resolution of 25 bits is compared as a suggested methodology for end users to identify sufficient ADC resolutions. Lastly, all data taken from a single combination of parameters (i.e., 23 years old, 515 nm, 0.03 VFM, Chebyshev Bandpass) is rescaled to range from 0.5 to 2.7 via MATLAB’s rescale function. This was performed to match the approximate range of values seen in the training data for feature extraction and machine learning.

#### 2.3.2. Feature Extraction

PPG feature extraction was performed in Python using in-house code and the heartPy library to extract the systolic peak and onset [43]. The feature extraction identifies 30 s of data wherein heartPy successfully extracts >90% of the expected number of systolic peaks and rejects fewer than 10% of the expected number of systolic peaks. Then, 38 features found in Table A1 are calculated with 3 fiducial points: the systolic onset determined from heartPy, the systolic peak determined from heartPy, and the dicrotic notch determined by identifying the “c” peak in the second derivative of the PPG waveform [44]. The second derivative is smoothed and scaled to enable easier detection for PPGs with less-visible dicrotic notches. Once all of the 38 features were extracted, we aimed to determine the effect of patient-specific and device factors on the value of these features. This was done by comparing the mean and standard deviation of each feature across each factor. Then, a colormap that shows the range of values for each feature across each factor was created to enable visual inspection of the data.

#### 2.3.3. Machine Learning Algorithms

As an example use-case of this framework, three separate models were trained to estimate blood pressure from PPG-derived features with MATLAB 2020b (The Mathworks, Natick, MA, USA). A Support Vector Machine (SVM) and Bagged Trees algorithm were developed with MATLAB’s Regression Learning Application and a Neural Network was developed with MATLAB’s Neural Network Fitting Application.

All models were trained with data from the publicly available and deidentified MIMIC III Matched Data Set [20]. From this database, 40,000 patient data samples from 2437 patients that included arterial blood pressure (ABP) measured from an invasive arterial line and PPG waveforms were identified via stratified random sampling and 17,517 data samples, 30 s in length, were passed to feature extraction. The systolic peak and diastolic peak within the ABP data over 30 s was used to determine the systolic and diastolic blood pressure, respectively. Extracted features were then pre-processed as follows: data with systolic/diastolic ranges greater than 20 mmHg/12 mmHg within the 30 s were discarded and statistical outliers (defined as samples with data beyond 3 standard deviations) were removed. PPG feature values were then rescaled from zero to one. These methods are derived from ISO-81060-2 standard [45].

The SVM was trained on 70% of the MIMIC-III data with 10-fold validation, and then tested on the remaining 30% of the data. All data were standardized together to have a mean of zero and standard deviation of one. A fine Gaussian kernel function with a kernel scale of 1.5 was used. The artificial neural network was trained on 70% of the data, tested on 15% of the data, and validated on the remaining 15% of the data. Levenberg-Marquardt backpropagation was used with 100 hidden layers and mean squared error as an error function. The bagged trees algorithm was trained on 70% of the MIMIC-III data with 10-fold cross-validation and tested on the remaining 30% of the data. After training the algorithms, features originating from the synthetic data were used to predict systolic and diastolic blood pressure and the standard deviations of these predictions were analyzed as a way to assess the sensitivity of these algorithms to patient and device specific factors. Lastly, the predicted blood pressure is analyzed via mean error and standard deviation. This decision was informed by “ISO-81060-2, Non-invasive Sphygmamonometers—Part 2: Clinical investigation of intermittent automated measurement type” [45].

## 3. Results and Discussion

### 3.1. PPG Signal Generator Verification

#### 3.1.1. Monte Carlo Model 

Figure 4 shows changes in PPG normalized AC and DC amplitudes with respect to wavelength, VFM, and age relative to the maximum amplitude (515 nm, 0.03 VFM, 23 years old for AC and 880 nm, 0.03 VFM, 23 years old for DC). Note that while AC and DC amplitudes are shown separately, the AC/DC ratio is another parameter discussed in literature to gauge PPG waveform signal quality. However, it was not used here since the smaller AC/DC values in dark skin tones require more photons to precisely and accurately resolve than what was used in this work. Since a target coefficient of variation of less than 10% was set as a target for simulation precision, the derived value AC/DC would have a coefficient of variation greater than 10% because it is a derived value consisting of the AC divided by the DC. This limitation is caused by the computational complexity of these simulations wherein it would take weeks to properly simulate this scenario. Differences between this work and others with respect to simulated conditions (age, wavelengths, etc.) and the lack of details in other work prevent exact comparisons from being made. However, it is valuable to demonstrate agreement of trends between the simulations presented here and data collected elsewhere.

Experimental and in silico research has been published exploring the relationship between PPG wavelength and amplitude. Moco et al. observed a decrease of PPG amplitude from 515 nm to 660 nm by 55% and then a subsequent 150% increase in signal from 660 nm to 880 nm [46]. We observed a similar trend, namely a 59% decrease in PPG amplitude from 515 nm to 600 nm and a 107% increase in signal from 660 nm to 880 nm. Factors such as skin thickness and device characteristics explain the absolute differences in results from 660 nm to 880 nm.

The second variable explored in Figure 4 is the impact of VFM on the PPG signal. Ajmal et al. used MC modeling to explore the impact of skin tone on the wrist, an anatomy with more melanin than the fingertip. Specifically, they explored the impact of Fitzpatrick Skin Tone I (set by Ajmal et al. to be 0.03 VFM) and Fitzpatrick Skin Tone VI (set by Ajmal et al. to be 0.42 VFM), on the AC/DC ratio of various commercial PPG-based heart rate monitors [47]. While error propagation in this study limits conclusions regarding the AC/DC ratio on dark skin tones, our results agree with those presented in the literature as AC/DC ratio does not change in 0.03 and 0.10 VFM [47]. Ajmal et al. presented that AC/DC ratio decrease as a function of skin tone ranged from less than 1% to approximately 15% depending on the wearable source/detector configuration. This study is different from the current effort, as Ajmal et al. performed their work at the wrist with different source to detector configurations.

The last variable shown in Figure 4 is age. The effect of age on PPG signal amplitude is largely unexplored, however previous work showed that at least two physiological changes create an effect: (1) decrease in skin thickness, and (2) decrease in vessel compliance. The former increases PPG signal amplitude and the later decreases PPG signal amplitude by reducing the change in blood volume caused by the cardiac cycle [11]. Both of these factors are included in determining the effect of age on PPG amplitude in Figure 4. In previous literature, the significant decrease in vessel compliance observed with age supports the trend observed in the results of a decrease in PPG amplitude [34]. To increase the accuracy of the model with respect to varying age, the epidermal layer should also decrease in thickness as age changes. This factor was not included in this work, as decreasing epidermal thickness would require increasing simulation resolution and to substantially increase the time required to collect results. It is hypothesized that decreased epidermal thickness would increase PPG signal, however it is unknown whether this effect would negate the decrease in signal caused by changes in vessel compliance. However, PPG amplitude is not the only change that age causes in PPG feature extraction, rather the main change is morphological [41].

#### 3.1.2. PPG Waveform Generator

The ability of the workflow to replicate PPG waveforms was assessed. Table 3 shows the Gaussian parameters used to generate waveforms across four ages. To analyze the ability of the gaussian combination method to replicate literature-sourced waveforms, median relative error was chosen as it is robust to large percent differences when the waveforms approach zero. In all cases, between 23 and 55 years old, median relative error was below 5% indicating strong ability to generate data derived from waveforms supplied by the end user or specified in this work. Qualitatively, as depicted in Figure 5, these waveforms also demonstrated well-studied morphological changes in PPG waveforms as a function of subject age: the dicrotic notch and diastolic peak become less noticeable as age increases [18]. These individual waveforms are repeated over a window of time to generate continuous data for signal processing, an example of this is shown in Figure 6.

### 3.2. Impact on PPG Morphology and Features

A simple ADC simulation component was included in the Device Algorithm Simulator to be able to study the minimum resolution needed to accurately resolve the PPG waveform features. Figure 7a,b shows a PPG waveform originating from a synthetic patient with an excitation wavelength of 515 nm, 0.30 VFM (b) or 0.03 VFM (a) and 23 years old (a) or 55 years old (b). This synthetic data was filtered through a bandpass Butterworth and then put through the ADC simulator. The lowest ADC resolution, shown in blue, has false features in both subfigures after the systolic peak that were artificially added by the filter in an attempt to process a digitized signal. Additionally, the systolic onset is greater and the systolic peak is less than their high-resolution counterparts. This waveform would be unable to undergo feature extraction, or would yield incorrect feature values if processed. However, as the resolution increases, the waveform regains its morphology. Figure 7c demonstrates the framework’s ability to analyze the impact of ADC resolution on waveform morphology. This data illustrates the relative percent difference between the PPG waveform at a given ADC resolution, compared to the same waveform at the high resolution of 25 bits which serves as a near-perfect waveform for this analysis. According to Figure 4, the synthetic data yielding the black curve is approximately 300× greater in amplitude than the magenta curve. This amplitude difference manifests itself in ADC resolution necessary to sufficiently resolve the signal. In order to obtain a signal <1% different than the high-resolution signal, a 10-bit resolution is necessary for the 23 year old, whereas the synthetic data yielding the magenta curve (55 year old) requires a 19-bit resolution. This functionality can also be used to evaluate appropriate LED intensity, as it is expected that increasing the LED intensity to yield a matching PPG amplitude across features would potentially yield equivalently resolved features, even though this action would similarly amplify noise and the DC component of the PPG waveform.

Figure 8, an abbreviated version of Figure A1, illustrates that by comparing values of a given feature within a patient-specific or device facto, we can study which factors modulate which features. Skin tone, or rather VFM from 0.03 to 0.30, and wavelength did not significantly impact any feature measured. This intuitively makes sense, as their representation within this framework is through changing optical properties of tissue and thus their impact is in the amplitude of the PPG waveform (shown in Figure 4). Dicrotic notch height, an amplitude-based feature in Figure 8, does not change as one of the aforementioned components of signal processing was data rescaling. There is very limited evidence suggesting that skin tone or wavelength may impact the PPG waveform, but such an effect would largely be caused by changes in optical properties of tissue leading to manipulating whether the PPG signal is predominately provided by superficial arterioles or deeper arteries [48]. However, in this work, where the simulated anatomy is primarily vascularized by superficial arterioles in the form of subungual arcades and their branches, this is not a likely outcome [49]. Filter methodology and age are shown to have significant impacts on extracted feature values. Specifically, “DivWidthTime” features decrease as age increases, and most “WidthTime” features increase as a function of age. As the target percentage for the WidthTime features decrease, the impact of age also decreases. For example, x75WidthTime increases from 0.16 to 0.23 for 23-year old’s to 55-year old’s, but x10WidthTime changes from 0.67 to 0.68 as age increases. This effect is inverted for “DivWidthTime” features as a higher target percentage changes less as a function of age. X75DivWidthTime changes from 2.53 to 2.12 as age increases and X10DivWidthTime changes from 4.68 to 3.26 as age increases. Signal filtering is shown to impact extracted feature values. The inverse Chebyshev bandpass has the least accurate performance, consistently leading to underestimated features values compared to the control data when the feature is related to timing. However, these waveforms assume that ADC resolution is sufficiently high. Identifying the appropriate ADC resolution is a key component of device design.

### 3.3. Blood Pressure Estimation from Synthetic PPG Features

A number of machine learning approaches have been studied for non-invasive blood pressure (BP) prediction from PPG [19,50,51]. This includes variations in features as well as variation in methodology/algorithms. As a test case of utilizing this framework to evaluate the robustness of trained algorithms against patient and device specific factors, a Support Vector Machine (SVM) algorithm, a Bagged Trees (BT) algorithm, and a Neural Network (NN) were developed. The algorithms were selected after analyzing commonly used techniques to predict blood pressure from PPG data in the literature. Similar to previous work, each model was trained using 70% of data extracted from the MIMIC-III dataset and tested on the remaining 30% of extracted data (15% for the NN, as 15% was withheld for validation) [52]. Table 4 displays the mean average error and standard deviation of error for the trained models on the MIMIC dataset in the first two rows, and the standard deviation of predicted values for each algorithm on the synthetic dataset in the bottom two rows. The bottom two rows show the standard deviation of prediction because the true blood pressure values in the synthetic data are unknown. Mean average error on the test data was found to be within ISO-81060-2 standards of <5 mmHg, and the standard deviation of the error was found to be greater than the ISO-81060-2 threshold of 8 mmHg for these example algorithms [45]. However, the purpose of this test case is to examine the variation of predicted blood pressures across synthetic data for each trained model, not to develop models for blood pressure prediction; thus we do not anticipate the standard deviation of error to impact conclusions derived within this work. While the true blood pressure of the synthetic data is unknown, the standard deviation of predictions from each algorithm can be used to assess sensitivity or insensitivity to the factors discussed in this work. It was found that the SVM had a low standard deviation of measurement of 0.10 mmHg and 0.12 mmHg for diastolic and systolic BP, respectively across synthetic data. In contrast, the NN had much greater standard deviations of prediction 14.53 mmHg and 9.44 mmHg for systolic and diastolic BP, respectively. Thus, this workflow was able to show that the systolic and diastolic outputs of the SVM models developed in this work were less sensitive to the diverse cohort of synthetic PPG signals, compared to the NN or Bagged Trees models.

## 4. Conclusions

Due to the number of anatomical, physiological, and device recording factors that can impact the morphology of a PPG waveform, it is important to have tools to enable the systematic assessment of hemodynamic measurement algorithms to these factors. We considered a framework and developed an initial software implementation towards this purpose. We demonstrated how this can be used to generate synthetic data specific to device characteristics and inclusive of patient-specific factors, one can systematically evaluate feature and algorithm robustness across a range of patient and device-specific characteristics. This type of framework can enable rapid development of algorithms and devices that aim to predict blood pressure, or potentially other hemodynamic measurements, from pulse wave analysis of the PPG waveform by combining in silico developmental tools to overcome individual limitations. Namely, this research includes both physiological and anatomical considerations in designing PPG-based medical devices and external sources of noise. Other frameworks that are Monte Carlo based explored changes in PPG amplitude caused by physiological factors, and frameworks that were Gaussian combination -based explored noise-based changes in the PPG waveform. The framework described here enables studying the impact of both categories on the PPG waveform. It also allows the understanding of how the effects may propagate through data recording and processing algorithms to the end clinical parameters by performing feature extraction on the simulated waveforms. This study presented the use of this framework for a single anatomy due to the computational power of the computers used for simulations. Increasing the computational power, and consequently the number of photons used in the Monte Carlo simulations would enable evaluation of derived parameters such as AC/DC, particularly for darker skin tones. The simulated PPG waveforms presented herein assume physiotypical subjects and are created/validated with data from literature, which limits control over the health of patients from which the data originated. It is anticipated that nonhealthy patients may present changes to the model that may include but are not limited to changes in skin thickness, optical properties of the skin layers, and vessel compliance. Thus, future work might include conducting studies to gather the parameters such as skin thickness used in this work that are controlled to important variables such as cardiovascular health. Additionally, some works used as reference have uneven gender distributions that might impact the results incorporated to build the simulations presented herein. Generalized signal processing was performed in this work that does not represent the full processing of any known device. Additionally, the research presented herein uses a single source/detector configuration as a case study and is not representative of any specific product configuration. Lastly, increased validation and uncertainty quantification would further enable functionalities such as including template PPG waveforms for patients of any age.

## Figures and Tables

**Figure 1 biosensors-12-00598-f001:**
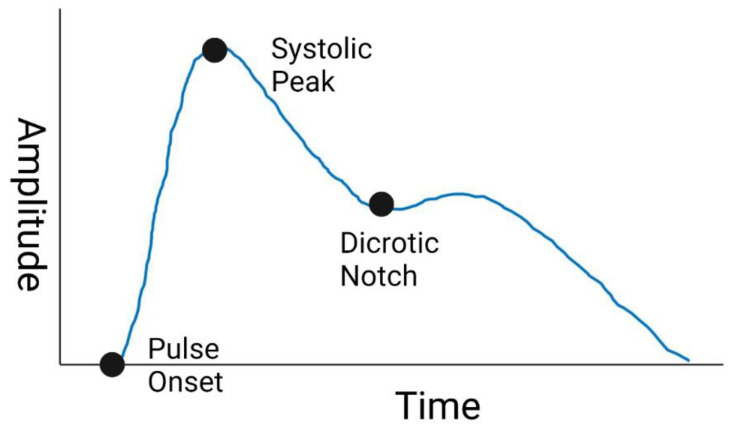
Sample PPG Waveform. Created with BioRender.com (accessed on 26 May 2022).

**Figure 2 biosensors-12-00598-f002:**
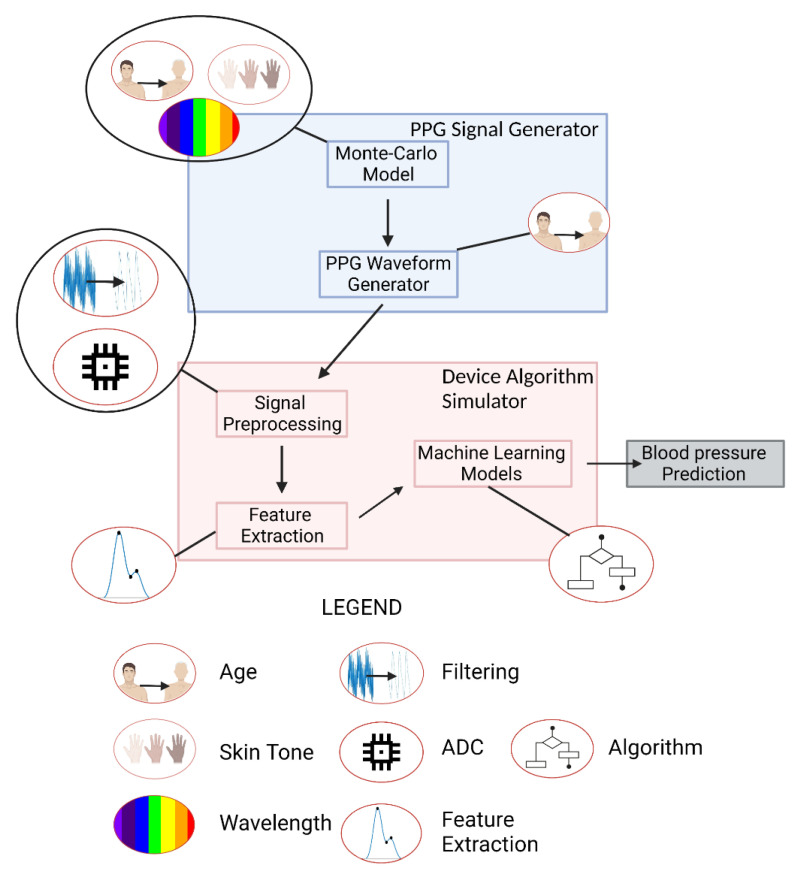
Computational Workflow. Created with Biorender.com (accessed on 26 May 2022).

**Figure 3 biosensors-12-00598-f003:**
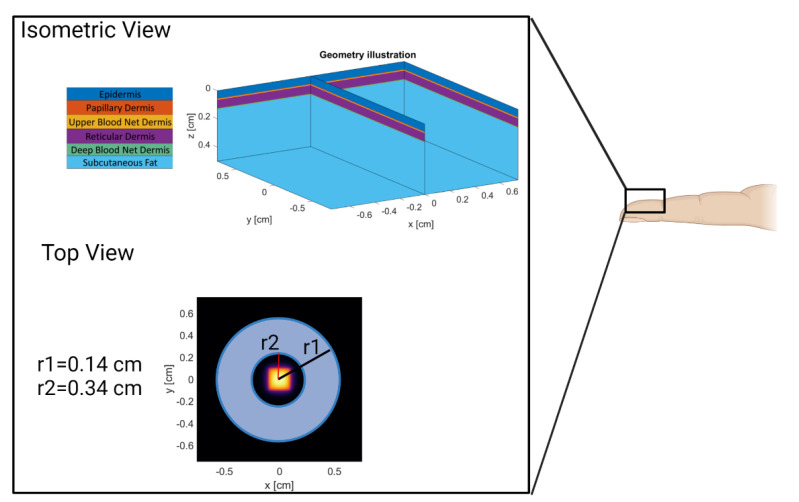
Model Geometry and Source/Detector Geometry. Created with Biorender.com (accessed on 26 May 2022).

**Figure 4 biosensors-12-00598-f004:**
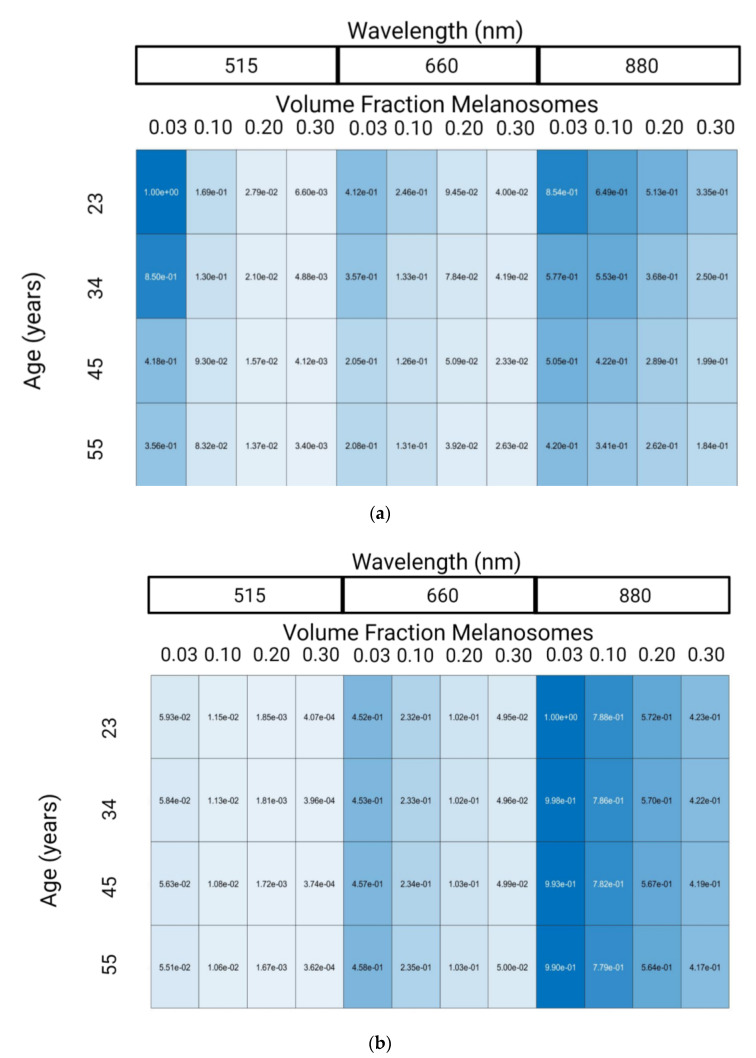
Normalized PPG AC amplitude (**a**) and PPG DC amplitude (**b**) from Monte-Carlo Simulation Results. Made with Biorender.com (accessed on 26 May 2022).

**Figure 5 biosensors-12-00598-f005:**
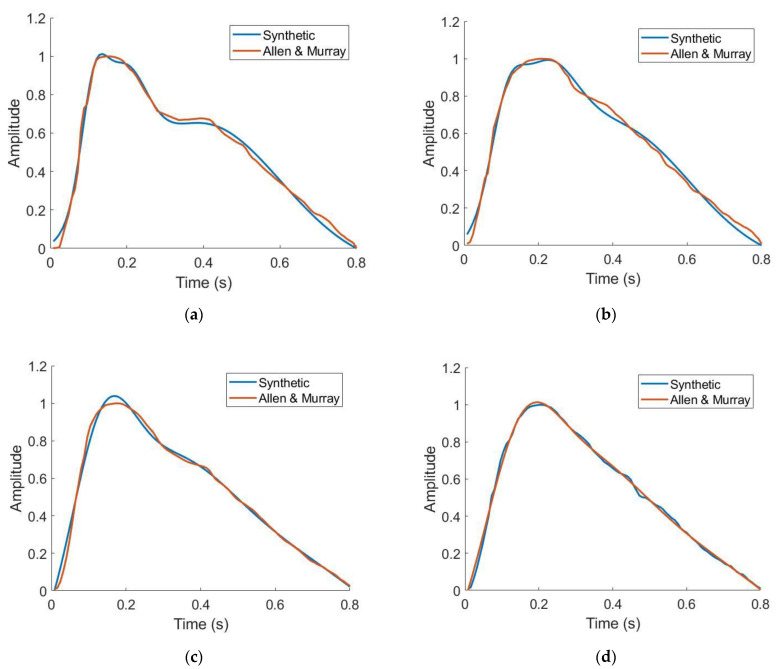
Synthetic vs. literature-derived PPG waveforms for 23 years old (**a**), 34 years old (**b**), 45 years old (**c**), and 55 years old (**d**) [41].

**Figure 6 biosensors-12-00598-f006:**
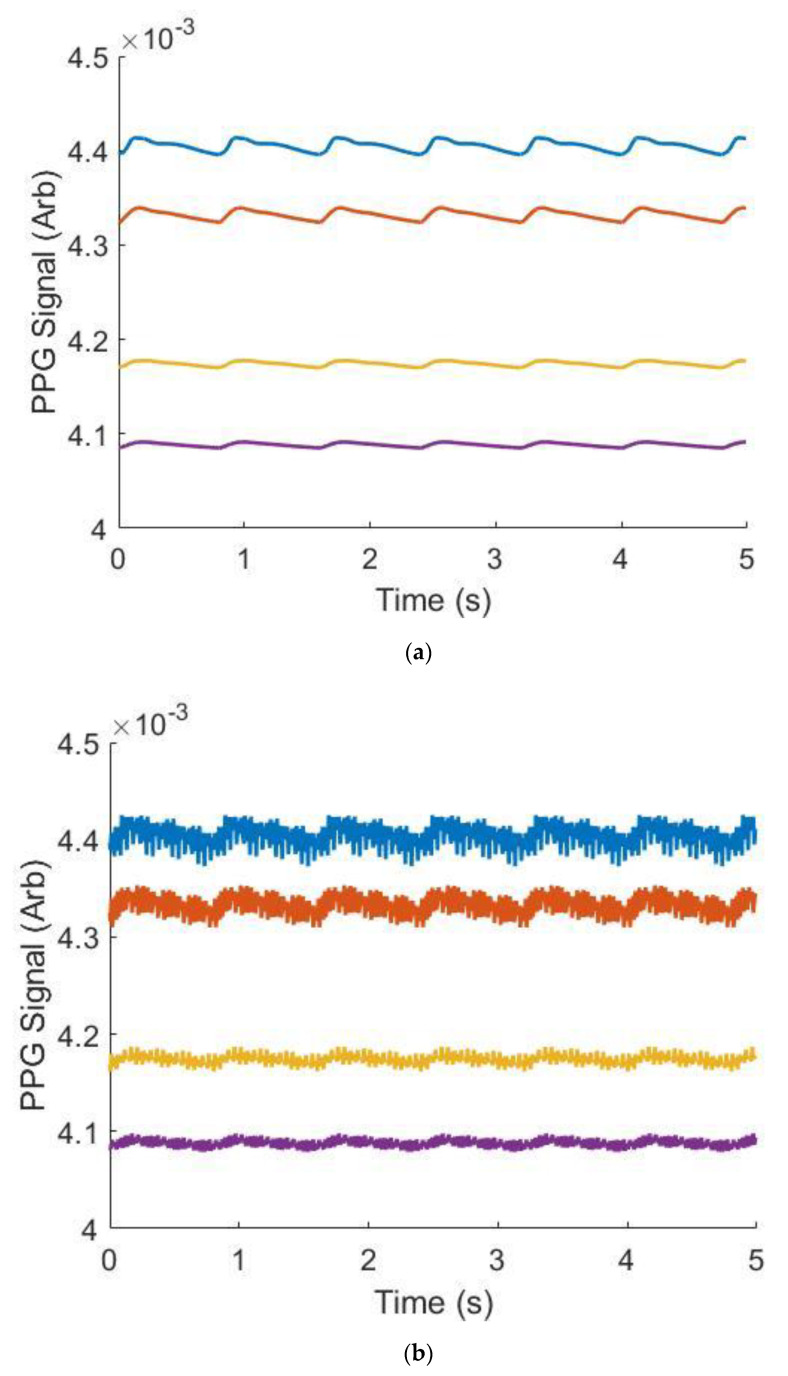
Example synthetic data across wavelengths and ages (VFM = 0.03, Wavelength = 515 nm), without (**a**) and with (**b**) noise. Blue is 23 years old, orange is 34 years old, yellow is 45 years old and purple is 55 years old.

**Figure 7 biosensors-12-00598-f007:**
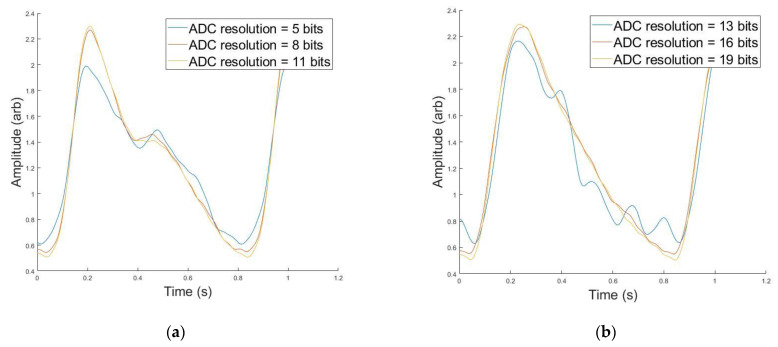
(**a**) PPG Morphology at Different ADC Resolutions (blue = 5 bits, red = 8 bits, orange = 11 bits) for 515 nm, 0.03 VFM, and 23 Years old. (**b**) PPG Morphology at Different ADC Resolutions (blue = 13 bits, red = 16 bits, orange = 19 bits) for 515 nm, 0.30 VFM, and 55 years old. (**c**) Median Percent Difference Between PPG Morphologies for Large and Small ADC Resolution for two simulated data.

**Figure 8 biosensors-12-00598-f008:**
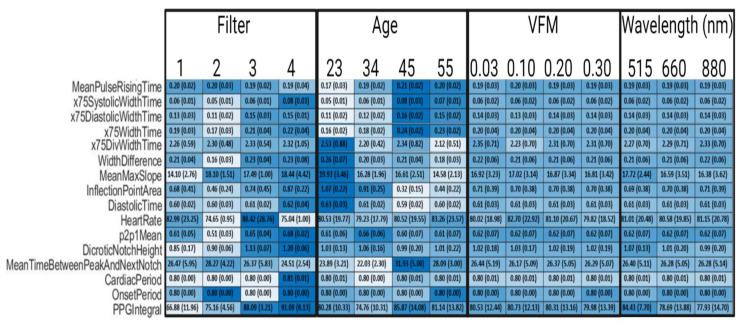
Selected Synthetic PPG Feature Values Across Parameters. Filter 1 is a 4th order bandpass Butterworth (0.1–0.7 Hz), Filter 2 is a 4th order bandpass Inverse Chebyshev (0.1–0.7 Hz), Filter 3 is a 4th order Low pass Inverse Chebyshev, and Filter 4 is control data that has no noise and no filters applied. Colormap was determined by normalizing each row. Standard deviations are in parentheses.

**Table 1 biosensors-12-00598-t001:** Common PPG Features.

Feature Name	Definition
Pulse Rise Time	Difference in time from pulse onset to systolic peak
Peak Amplitude	Difference in signal amplitude between systolic peak and onset (AC component)
“X”% Systolic Width	Difference in time between “Y” and the systolic peak, where “Y” is the time at which “X”% of the peak amplitude is achieved before the systolic peak [19]
“X”% Diastolic Width	Difference in time between “Y” and the systolic peak, where “Y” is the time at which “X”% of the peak amplitude is achieved after the systolic peak [19]
Inflection Point Area	The ratio a2/a1, where a2 is the area under the PPG waveform from the dicrotic notch to the next onset and a1 is the area under the PPG waveform from the onset to the dicrotic notch [18]
Pulse Rate	The number of systolic peaks observed over 60 s

**Table 2 biosensors-12-00598-t002:** Model Optical and Geometric Properties.

	Age (Years)					Wavelength (nm)					
	23	34.4	44.8	55		515		660		880	
	Thickness (mm)/Starting Depth (mm)					µ_a_ (cm^−1^)	µ_s_ (cm^−1^)	µ_a_ (cm^−1^)	µ_s_ (cm^−1^)	µ_a_ (cm^−1^)	µ_s_ (cm^−1^)
**Epidermis**	0.55/0.00	0.55/0.00	0.55/0.00	0.55/0.00	**(0.03/0.10/0.20/0.30 VFM)**	1.96/6.28/12.43/18.58	388.35	0.86/2.75/5.44/8.13	303.03	0.33/1.06/2.09/3.13	227.27
**Papillary Dermis**	0.075/0.55	0.071/0.55	0.067/0.55	0.064/0.55	**Rest**	1.2166	389.99	0.5249	208.65	0.2344	118.94
					**Pulsed**	1.2202		0.5250		0.2346	
**Upper Blood Net Dermis**	0.04/0.63	0.038/0.62	0.036/0.62	0.033/0.61	**Rest**	1.5328	389.99	0.5398	208.65	0.2546	118.94
					**Pulsed**	1.5593		0.5410		0.2558	
**Reticular Dermis**	0.75/0.67	0.71/0.66	0.67/0.65	0.64/0.65	**Rest**	1.2167	389.99	0.5256	208.65	0.2456	118.94
					**Pulsed**	1.2202		0.5257		0.2458	
**Deep Blood Net Dermis**	0.05/1.42	0.05/1.37	0.04/1.33	0.04/1.28	**Rest**	1.2896	389.99	0.5288	208.65	0.2462	118.94
					**Pulsed**	1.2985		0.5292		0.2466	
**Subcutaneous Tissue**	2.00/1.47	2.00/1.42	2.00/1.37	2.00/1.33	n/a	6.0798	336.18	0.2827	249.74	0.3195	191.53

**Table 3 biosensors-12-00598-t003:** PPG Waveform Gaussian Parameters and Median Relative Error for GRG nonlinear solver.

Age (Years)	Gaussian 1 Parameters (a1, b1, c1)	Gaussian 2 Parameters (a2, b2, c2)	Gaussian 3 Parameters (a3, b3, c3)	Median Relative Error (%)
23	0.57,0.19,0.09	0.47,0.11,0.05	0.77,0.39,0.30	3.58
34.4	0.80,0.28,0.25	0.77,0.59,0.44	0.74,0.13,0.11	2.12
44.8	0.59,0.21,0.12	0.38,0.11,0.06	0.75,0.40,0.29	4.14
55.0	0.77,0.28,0.25	0.67,0.14,0.13	0.79,0.58,0.44	1.79

**Table 4 biosensors-12-00598-t004:** Algorithm Mean Error and Standard Deviation.

	Support Vector Machine		Bagged Trees		Neural Network	
	Mean Error (mmHg)	Standard Deviation (mmHg)	Mean Error (mmHg)	Standard Deviation (mmHg)	Mean Error (mmHg)	Standard Deviation (mmHg)
Systolic	0.55	11.56	−0.02	12.4	−0.36	15.53
Diastolic	−0.72	8.24	−0.11	8.63	−0.16	10.75
Systolic-Synthetic	N/A	0.12	N/A	3.46	N/A	14.53
Diastolic-Synthetic	N/A	0.10	N/A	4.34	N/A	9.44

## Data Availability

Software and Data used in this work will be provided by the authors upon reasonable request.

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
