# Peer review of "A Computational Modeling and Simulation Workflow to Investigate the Impact of Patient-Specific and Device Factors on Hemodynamic Measurements from Non-Invasive Photoplethysmography"

_biosensors, 2022, doi:10.3390/bios12080598_

Round 1
Reviewer 1 Report
This paper presents a computational simulation workflow to study the impact of patient-specific and device factors from PPG hemodynamic measurement. Nevertheless, after including additional information, its publication should be considered, making it more convincing that the simulation method would work with good results. Generally, they only show and explain the simple results without sound analysis, discussion, or trustworthy guidance. The authors should present a clear conclusion regarding the impact of patient-specific and device factors from PPG measurements to enhance the study's motivation and purpose. The paper is not well prepared to read and understand and has many typos and unclear/low-resolution figures.
- The authors use the combination of MC and Gaussian combination, etc. Is it just a method to combine previous methods? What kinds of merits are compared to the conventional way?
- The authors introduced the background on the estimation of blood pressure from PPG in the introduction. How could this simulation work correlate/contribute to predicting blood pressure? Please, explain the points according to your methods.
- The authors showed the results of predicting blood pressure using machine learning. And SVM, bagged trees, and neural networks were selected as algorithms for prediction. SVM and bagged trees can be machine learning algorithms. However, neural networks can be deep learning algorithms. Any reasons to use both machine learning and deep learning algorithms?
- When applying the learning algorithm, 70% of the total data was used as the training set. Any reason to use the training set ratio as 70%, not 80%?
- Table 5 shows accuracy and precision, but the actual average value and standard deviation of the predicted blood pressure are shown. Is it a typo, or is it a result to evaluate the learning algorithm's performance, such as accuracy and precision?
- When evaluating a learning algorithm's performance, several indicators can show the performance, such as the F1 score and AUC score, in addition to accuracy. Any reasons to choose only accuracy and precision? If possible, include others.
- All standard deviations of systolic predicted by the learning algorithm exceed 10mmHg, and the standard deviation of diastolic also exceeds 8mmHg. It is a difference of about 10% based on normal blood pressure. The authors should explain whether this level of accuracy is reasonable or not.
- In line 190, MCmatlab was mentioned. Is there other library or valuable resources that can be used in the research? Also, is there any reason to choose this library?
- In lines 209~215, the input photodiode and LED size were explained, and the authors used the set-ups similar to the Apple Watch LED/PD configuration. What happens in the result if you use other LED/PD configurations?
- In table 2 and lines 239 ~ 240, there were model and geometric properties. The Epidermis thickness is the same in all cohorts. Your models have the same epidermis thickness. If the thickness is changed, how can the results be changed?
Also, there are some minor comments.
1. Figure 2 was hard to know what each small figure meant. Also, what does the figure connected to the PPG waveform generator mean? Where did you explain it in the manuscript? In figure 3, please describe the upper-right figure.
2. It was hard to understand when you explained the figure with left, right, upper and bottom. It would be better to know if each subfigure is notated as (a), (b), (c).
3. In table 2, because of the table format, it was hard to know what critical data is and hard to match column and row.
4. There are typos or format errors. the "incporate" in line 180, "5E8" in line 224 and "1E11" in line 225.
Reviewer 2 Report
Please see attached file for comments

Round 2
Reviewer 1 Report
The authors answer my review points faithfully.
I do not have any questions.
Reviewer 2 Report
This revised version shows improvement in the presentation of the results as well as clarity of content. The reviewer's recommendation is that the changes made are sufficient enough to merit this paper's publication